# Radiation exposure and clinical outcome in patients undergoing percutaneous intradiscal ozone therapy for disc herniation: Fluoroscopic versus conventional CT guidance

**Francesco Somma**[1]*, **Vincenzo D'Agostino**[1], **Alberto Negro**[1], **Valeria Piscitelli**[1], **Stefania Tamburrini**[2], **Carmine Sicignano**[1], **Fabrizio Fasano**[1], **Silvio Peluso**[3], **Alessandro Villa**[4], **Gianvito Pace**[1], **Giuseppe Sarti**[2], **Giuseppe Maria Ernesto La Tessa**[1], **Giovanna Pezzullo**[5], **Gianluca Gatta**[5], **Ferdinando Caranci**[5]

**1** UOC Neuroradiologia, Ospedale del Mare, ASL NA 1 Centro, Napoli, Italy, **2** UOC Radiologia, Ospedale del Mare, ASL NA 1 Centro, Napoli, Italy, **3** UOC Neurologia, Ospedale del Mare, ASL NA 1 Centro, Napoli, Italy, **4** UOC Neurochirurgia, Ospedale del Mare, ASL NA 1 Centro, Napoli, Italy, **5** Dipartimento Medicina di Precisione, Università "Vanvitelli", Napoli, Italy

* fra1585@hotmail.it

## Abstract

### Purpose

To compare technical success, clinical success, complications and radiation dose for percutaneous intradiscal ozone therapy in patients with lumbar disc herniation using fluoroscopic guidance versus conventional computed tomography (CT) guidance.

### Materials and methods

Between March 2018and March 2021, 124consecutive percutaneous intradiscal ozone therapies wereperformedon111 patients with low back pain (LBP) and/or sciatic pain due to lumbar disc herniation, using fluoroscopic or conventional CT guidance, respectively in 53 and 58 herniated lumbar discs, with at least 1-month follow up. Dose area product (DAP) and dose length product (DLP) were recorded respectively for fluoroscopy and CT, and converted to effective dose (ED).

### Results

Fluoroscopic and CT groups were similar in terms of patient age (p-value 0.39), patient weight (p-value 0.49) and pre-procedure Oswestry Disability Index (ODI, p-value 0.94). Technical success was achieved in all cases. Clinical success was obtained in 83.02% (44/53) patients in fluoroscopic group and 79.31% (46/58) in CT group. Mean DAP was 11.63Gy*cm$^2$ (range 5.42–21.61). Mean DLP was 632.49mGy-cm (range 151.51–1699). ED was significantly lower in the fluoroscopic group compared toCT group (0.34 vs. 5.53mSv, p = 0.0119). No major complication was registered. Minor complications were observed in 4 cases (2 in fluoroscopic group; 2 in CT group).

**Data Availability Statement:** All data are contained within the paper.

**Funding:** The author(s) received no specific funding for this work.

**Competing interests:** The authors have declared that no competing interests exist.

## Conclusions

Compared to conventional CT guidance, fluoroscopic guidance for percutaneous intradiscal ozone therapy in patients with lumbar disc herniation shows similar technical and clinical success rates, with lower radiation dose. This technique helps sparing dose exposure to patients.

## Introduction

Low back pain (LBP) is one of the most common and important clinical, social, economic, and public health problems affecting the human population worldwide [1]. Around 70% of adults suffer from LBP during their lifetime with different symptom severity, and half of them have LBP associated with sciatic symptoms [2, 3]. The mechanism of lumbar pain is still not fully understood and seems to be multifactorial, thus considering pain as the result of compression, irritation, and chemical inflammation of peripheral nerves surrounding intervertebral discs [4, 5]. In particular, it is believed that disc herniation produces a mechanical stress on the adjacent nerve root causing the release of inflammatory cytokines, which makes the nerve root oversensitive to mechanical compression itself [6]. Magnetic Resonance (MR) is the method of choice for the diagnosis of radicular disc compression, allowing the analysis of all structures involved in LBP [7]. Possible treatments include physical and/or drug therapy, massage traction, minimally invasive therapy, and surgical therapy [8, 9]. All treatments have the same rationale that consists in reducing disc volume and nerve root compression. The natural history of disc herniation is often favorable but symptoms may persist for long periods of time in around 37% of patients [1].

To avoid or delay open surgery, percutaneous minimally invasive techniques have been developed in recent years. Among these, the percutaneous injection of intradiscal oxygen-ozone (O2-O3) has found a wide application because of its clinical success rates and low procedural costs. Fluoroscopic or conventional Computer Tomography (CT) guidance is used to allow disc and/or foraminal access, with different dose exposure. So far, the choice of the radiological guidance has be driven mainly by the operator preference and by the technique availability. Nowadays, the strong attention of the scientific community on radiation protection forces to consider this parameter, too.

Therefore, aim of this study is to assess technical success, clinical success, safety and radiation dose for percutaneous intradiscal ozone therapy in patients with lumbar disc herniation using fluoroscopic versus CT guidance.

## Material and methods

### Ethics statements

Institutional Review Board approval for this study was waived because the studied procedure is ordinarily performed in our institution, and is not considered experimental. Appropriate written informed consent was collected before every procedure. All data were retrospectively collected. The authors declare that they have no conflict of interest. No funding was received to support this study.

### Patients population

Between March 2018 and March 2021, 124 percutaneous intradiscal ozone therapies were performed on 111 patients with LBP and/or sciatica due to lumbar disc herniation, diagnosed on

MRI and/or CT. In 12 patients, two disc herniations at different levels were treated in the same session. Nine patients underwent a second procedure within 45 days after the first intradiscal ozone injection.With regard to the guidance, the choice of fluoroscopy versus CT was mainly due to the availability of the CT room (also used for urgencies) and the angiographic room (also used for neurovascular interventions). All cases were retrospectively reviewed. Inclusion criteria were: LBP due to lumbar disc herniation and lack of response to medicaments. Exclusion criteria were: presence of vertebral malignancies and previous lumbar surgery.

## Ozone generator and administration technique

Ozone was generated using a commercially available oxygen/ozone generator (Medical 99 IR, Multiossigen s.r.l., Gorle, Italy), with the concentration of the oxygen/ozone mixture constantly displayed on the equipment monitor. The operation was performed under sterile conditions, with lumbar region thoroughly disinfected and sterile drapes applied. No local anesthesia was administered in the puncture site before the procedure. With patients in prone position, ozone was injected through a sterile 22-gauge 15cm-long Chiba needle (Cook Medical, Bloomington, IN) placed in the centre of the disc with percutaneous approach under imaging guidance. Using a polypropylene 10mL syringe, 5mL of ozone/oxygen mixture at 30μg/mL concentration rate was injected into the disc. Using a different syringe, 5mL of ozone/oxygen mixture at the same concentration rate was injected into the soft paravertebral tissues just outside the disc annulus around the nerve root, followed by the injection of 2mL solution containing 1mL of Depo-Medrol 40mg/mL and 1mL of Lidocaine-Hydrochloride 20mg/mL. Oral antibiotic therapy (generally Lovofloxacin 500mg) was prescribed for 5 days, starting from the day before the procedure. In case of exacerbated pain, the additional taking of paracetamol was recommended. Patients did not receive any other pre-operation or post-operation medications. After the chemonucleolysis procedure, all patients were discharged within two hours after the treatment and asked to avoid strenuous activities for 2 days.

## Imaging guidance

Both fluoroscopy and CT allow a good visualization of bony structures as well as the needle and its tip. CT is superior in the study of soft tissues and gas location after ozone injection. On the other hand, fluoroscopy reduces the procedural time but implies some operator exposure to radiations. So far, no significant difference in technical or clinical success has been described in patients treated with percutaneous intradiscal ozone injection, and the choice of the technique depends on the operator preference and/or the department organization.

## Fluoroscopy-Guided procedure

The patient lays in prone position on the Digital Subtraction Angiography (DSA) operating table (Allura X per FD20, Philips Medical, Eindoven, NL). At first, the lateral projection is performed and alignment of the end plates of the involved disc space is obtained through cranial and caudal angulations of the C-arm to clearly open the disc space. Then, the C-arm is rotated at an angle of 30˚–40˚, so that the facet joint superimposed on the posterior third of the disc space, can produce the so-called 'Scotty dog' appearance. In general, the puncture site is 7-10cmaway from the vertebral spinous process line. The needle is used to puncture the center of the disc under DSA, with an angle of 30–40˚, with insertion along the 'security triangle' through the posterior-lateral pathway. In the oblique projection, the needle is inserted just anterior and lateral to the superior articular process of the inferior vertebra ('Scotty dog' ear) at the direction of the X-ray beam. Once the position of the needle tip is confirmed in the

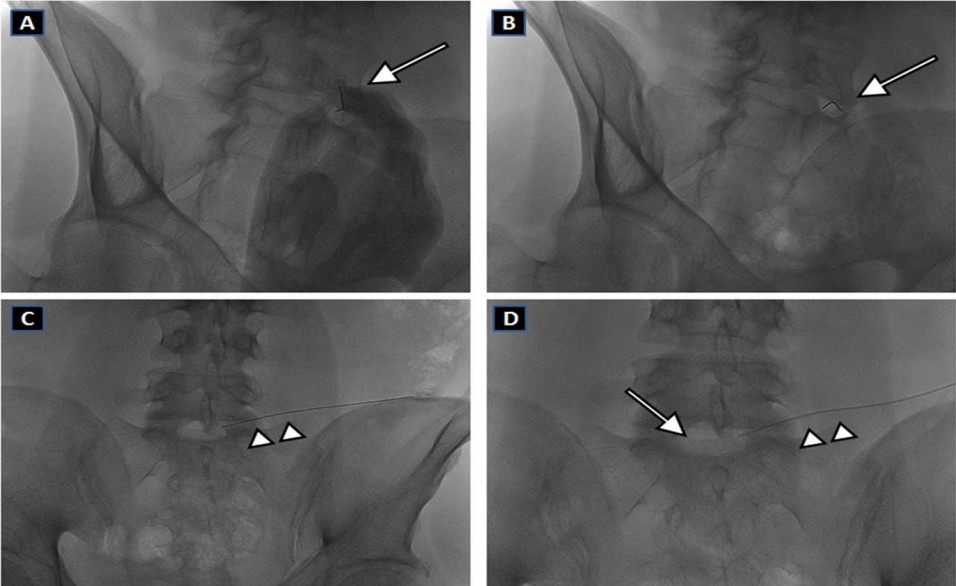

**Fig 1. Fluoroscopy-guided intradiscal ozone therapy of right-sided L5-S1 disc herniation.** Prone decubitus. (A, B) Oblique fluoroscopic images showing the Chiba needle (arrow) during insertion and advancement. (C) Antero-posterior fluoroscopic view, obtained before oxygen-ozone (O2-O3) mixture injection, showing the needle (arrowhead) tip inside the center of the concerned disc space. (D) Antero-posterior fluoroscopic view, obtained after oxygen-ozone (O2-O3) mixture injection, showing the injected gas as a faint opacity within the disc space.

center of the disc by both posterior-anterior and lateral fluoroscopy images (as shown in Fig 1), the administration is started.

## CT-Guided procedure

The patient lays in prone position on the CT sliding table (Acquilon Prime SP, Toshiba Medical Systems Europe, Zoetermeer, NL). CT sections for intervention planning are obtained in 3-mm slice thickness, parallel to the edge of the vertebra. Following, a CT-guided discography is performed, according to the orientation and angle determined by the CT scan. Under CT guidance the needle is advanced through the soft tissues to the nucleus pulposus, with a lateral approach by the same side of major pain. When the needle tip is well positioned (as shown in Fig 2), the administration is started. In case of puncture site at the level of L5-S1, a pillow is used to increase the lumbosacral angle. A post-procedural CT scan is performed to confirm the distribution of gas after the injection.

## Outcome measures

Technical success was defined as the correct placement of the needle tip in the herniated disc. Patients underwent a 1-month follow-up to determine procedure effectiveness. Oswestry Low Back Pain Disability Questionnaire [10] was administered to all patients on the day of the procedure and after 1 month. ODI is a percentage score, calculated on the base of a 10-item questionnaire, one item for pain and the others to assess the pain impact on daily life activities (personal care, lifting, walking, sitting, standing, sleeping, sexual life if applicable, social life, traveling). The response to treatment was considered binary in accordance with literature recommendations [11, 12]: it was categorized as successful (responders) in case of reduction in the preoperative ODI values of at least 30% during follow-up; unsuccessful (non-responders) if otherwise. Patients unable to return the questionnaire were questioned on telephone.

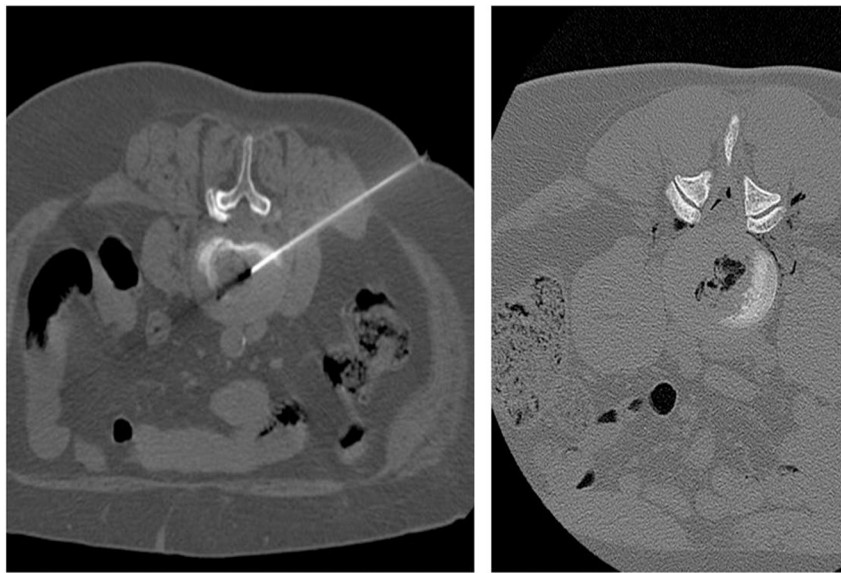

**Fig 2. CT-guided intradiscal ozone therapy of L4-L5 disc herniation.** Prone decubitus. On the right, axial CT scan image, obtained with bone tissues reconstruction algorithm, showing a lateral percutaneous approach with the needle tip placed into the nucleus pulposus. On the left, axial CT scan image, obtained after the procedure, showing ozone gas distribution in the disc and in the paravertebral soft tissues.

Radiation dose for each procedure was estimated by collecting dose area product (DAP) and dose length product (DLP) and converting them to effective dose (ED) for comparison. The following published conversion factors were considered: a) for DAP to ED conversion, established conversion coefficients based on phantom models were used, ranging from 0.0034 to 0.0101mSv/mGy*cm [2, 13], considering patient age and area of body scanned [14, 15]; b) for DLP to ED conversion, established conversion coefficients (k-factors) based on phantom models were used, ranging from 0.0003 to 0.0271mSv/mGy*cm [13], considering patient age and area of body scanned.

## Statistical analysis

The statistical analysis was performed with Matlab statistical toolbox version 2008 (Math-Works, Natick, MA, USA) for Windows at 32 bit. All data were analyzed as numbers and percentage for qualitative variables and mean and range for quantitative variables. Chi-squared test for qualitative variables and with Student's t test for quantitative variables were used to compare the groups' characteristics. Tests with p-value ($p$) $< 0.05$ were considered significant.

## Results

Tables 1 and 2 show the demographics of the patient population and the disc herniation characteristics, respectively.

Between March 2018 and March 2021, a total of 124 percutaneous intradiscal ozone therapies were performed on 111 patients aging from 22 to 78 years (mean age: 53.47 years; standard deviation: 14.90) for disc herniation in our interventional department and were retrospectively reviewed. Sixty-four of them (57.66%) were male, 47 of the 111 (42.34%) were female.

Fifty-tree patients were treated using fluoroscopic guidance and the remaining 58 were treated using CT guidance. These groups were similar in terms of patient age (p-value 0.39)

**Table 1. Demographics of the patient population.**

| Parameters | Overall (n = 111) | Fluoroscopy (n = 53) | Conventional CT (n = 58) | p-value |
|---|---|---|---|---|
| *Age* | | | | |
| (years), mean ± SD | 53.47 ± 14.90 | 51.65 ± 14.41 | 55.10 ± 15.39 | **0.39** |
| *Weight* | | | | |
| (kg), mean ± SD | 80.02 ± 12.92 | 78.73 ± 11.10 | 81.17 ± 14.46 | **0.49** |
| *Gender* (%) | | | | |
| Male | 64/111 (57.66) | | | |
| Female | 47/111 (42.34) | | | |
| *Main complain* (%) | | | | |
| Sciatica | 59/111 (53.15) | | | |
| Lumbalgia | 12/111 (10.81) | | | |
| Lumbalgia and Sciatica | 40/111 (36.04) | | | |
| *Disability symptoms* (%) | | | | |
| Mild | 15/111 (13.51) | | | |
| Moderate | 61/111 (54.95) | | | |
| Severe | 31/111 (27.93) | | | |
| Extreme | 4/111 (3.61) | | | |
| *Pain duration before treatment* (months), mean ± SD | 4.27 ± 2.44 | | | |
| *Pre-procedure ODI* (months), mean ± SD | 37.24 ± 14.34 | 37.38 ± 15.48 | 37.10 ± 13.52 | **0.94** |

SD, standard deviation.

ODI, Oswestry Disability Index.

and weight (p-value 0.49), intersomatic level treated (L4-L5, L5-S1, L4-L5 combined with L5-S1, p-value 0.13) and side of the disc herniation (p-value 0.36). The average age for each group was 51.65±14.41 years (range: 22–78 years) for fluoroscopic guidance group and 55.10 ±15.39years (range: 26–78 years) for CT group. The average weight was 80.02±12.92 kg (range: 55–99 kg) for fluoroscopic guidance group and 78.73±11.10kg (range: 48–115 years) for CT group. The mean duration of symptoms before treatment was overall 4.27 months (range:

**Table 2. Disc herniation characteristics sorted by guidance technique (Fluoroscopy, conventional CT).**

| | Fluoroscopy (n = 53, %) | Conventional CT (n = 58, %) | Total (n = 111, %) | p-value |
|---|---|---|---|---|
| **Level of disc herniation** | | | | **0.13** |
| L4-L5 | 10 (18.87) | 13 (24.53) | 23 (40.35) | |
| L5-S1 | 8 (15.09) | 9 (19.98) | 17 (32.08) | |
| L4-S1 and L5-S1 | 1 (1.89) | 2 (3.77) | 3 (5.66) | |
| **Side of the Lesion** | | | | **0.36** |
| Left | 13 (24.53) | 17 (32.08) | 30 (56.60) | |
| Right | 10 (18.87) | 13 (24.53) | 23 (43.40) | |
| **Disc lesion morphology** | | | | **0.22** |
| Bulges | 5 | 7 | 12 (10.81) | |
| Protrusion | 26 | 28 | 54 (48.65) | |
| Extrusion | 22 | 23 | 45 (40.54) | |

L, lumbar vertebra.

S, sacral vertebra.

SD, standard deviation.

**Table 3. Outcomes sorted by guidance technique (Fluoroscopy, conventional CT).**

|  | Fluoroscopy (n = 53, %) | Conventional CT (n = 58, %) | Total (n = 111, %) | p-value |
|---|---|---|---|---|
| Technical success | 53/53 (100%) | 58/58 (100%) | 111/111 (100%) | **0.17** |
| Pre-procedure ODI | 37.24 ± 14.34 | 37.38 ± 15.48 | 37.10 ± 13.52 | **0.94** |
| Post-procedure ODI | 14.31 ± 13.18 | 16.41 ± 10.08 | 15.42 ± 11.58 | **0.51** |
| ODI mean reduction | 23.08 ± 11.22 | 20.69 ± 10.20 | 21.82 ± 10.66 | **0.41** |
| Clinical success | 44/53 (83.02%) | 46/58 (79.31%) | 90/111 (81.09%) | **0.16** |
| Major complications | 0 | 0 | 0 | |
| Minor complications | 2/53 (3.77%) | 2/58 (3.45%) | 4/53 (3.61%) | |
| DAP (Gy*cm2) | 11.63 (5.42–21.61) | - | - | |
| DLP (mGy-cm) | - | 632.49 (151.51–1699) | - | |
| Effective dose (mSv) | 1.01 (0.09–1.99) | 9.48 (2.28–25.49) | - | **<0.001** |

Gy, Grey.

mSv, milliSievert.

ODI, Oswestry Disability Index.

1–9months, standard deviation: 2.44). Most patients presented with sciatica (59/111, 53.15%), 12 (10.81%) mainly suffered of lumbalgia, and the remaining 40 (36.04%) complained both lumbalgia and sciatica. Symptoms were mild in 15 patients (13.51%), moderate in 61 (54.95%), severe in 31 (27.93%) and extreme in 4 (3.61%).

Outcomes are showed in Table 3.

Technical success of 100% was registered in both groups. According to Oswestry Low Back Pain Disability Questionnaire, the mean pre-procedure ODI score was 37.24±14.34for the fluoroscopy group and 37.38±15.48 for the CT group, with no significant difference between them. The mean post-procedure ODI score was 14.31±13.18 for the fluoroscopy group and 16.41±10.08 for the CT group, with no significant difference between them. The mean reduction and percentages of symptoms improvement for ODI at 1-month follow up was 23.08 ±11.22 (61.57%) and 20.69±10.20 (56.10%) for the fluoroscopy group and the CT group, respectively. One-months ODI evaluation showed a significant improvement of disability symptoms in 83% of patients treated using fluoroscopy and in 79.3% of patients treated using CT guidance. No statistical significant difference was found between these groups (ODI mean reduction, p-value = 0.41). Overall, we registered a significant relief of pertinent symptoms in 90/111 (81.1%) patients, with a mean decrease of ODI score of 21.82±10.66 (mean overall pre-procedure, 37.10±13.52; mean overall 1-month follow up, 15.42±11.58).

In case of fluoroscopic guidance, the mean fluoroscopy time per procedure was 3.1min (range 2.3–7.7) with a mean DAP of 11.63Gy*cm2 (range 5.42–21.61; standard deviation 2.98). In case of CT guidance procedures a mean DLP of 632.49mGy-cm (range 151.5–1699; standard deviation 475.03) was registered. Estimated effective radiation doses were significantly lower in the fluoroscopy group compared to the CT group (1.01 vs. 9.48mSv, p < 0.001).

The procedure was overall well-tolerated. In this study, there was no major complication such as disc infection, nerve or vascular injury, and pneumocephalus. Minor complications were described in 2 patients for each group (1 had post-procedure syncope; 3 complained of low back pain exacerbation within the first 24 hours after the treatment).

## Discussion

Mini-invasive techniques are more and more used for many pathologies, both oncologic and non-oncologic [16–20]. Oxygen-ozone chemonucleolysis is a safe and minimally invasive disc

therapy, commonly used worldwide for the treatment of disc herniation since the 1990s [16]. It is based on the administration of medical ozone (O2-O3 mixture)at non-toxic concentrations (5–40µg of O3 per ml of oxygen),prepared through conversion of pure oxygen (O2) into ozone (O3) using generators that can modify ozone concentration as required [21, 22].

In order to insure delivery of O2-O3 right into the disc, the injection is imaging–guided: radiological guidance enables needle trajectory planning and monitoring. Procedures are generally performed using fluoroscopic or CT guidance. The first modality grants a simple, rapid, real-time monitoring of the advancing needle but implies some X-ray exposure for the operator. Conversely, CT guidance provides a precise non-real time control of needle insertion but avoids radiation exposure to the operator. In both cases, patients receive a single-session treatment of intradiscal ozone injection with concomitant periradicular infiltration of the oxygen-ozonemixture, steroid and local anesthetic, as the combined injection of these materials has proved to reach a better outcome in comparison with the use of ozone or steroid alone [12, 23]. In any case, radiation exposure to patients is necessary to reach a proper injection site for the needle tip.

Recently, "Image Gently" and "Step Lightly" campaigns well highlighted the importance of minimizing radiation exposure for patient [24, 25]. Also due to the increasing number of medico-legal disputes on this topic, many countries are imposing by law to display the radiation dose details in every radiological report, just below the usual findings communication [26]. Therefore, always more and more attention is paid on the application of 'as low as reasonable achievable' (ALARA) principles for limiting ionizing radiation exposure.

Compared to CT guidance, the use of fluoroscopy allows a decreased radiation exposure to patients undergoing percutaneous intradiscal ozone therapy for disc herniation, as showed by results in our series. This fact was proved through the direct comparison of radiation dose after the conversion of standard recorded radiation output (DAP for fluoroscopic guidance; DLP for CT guidance) to ED, which is a weighted average of organ doses modified on the base of tissue-weighting factors recommended by the International Commission on Radiological Protection (ICRP) Publication 160 [24]. ED is generally used to account for the biological effects of radiation, thus representing an effective tool for the comparison of radiation dose due to different imaging modalities.

An advanced research on *pubmed* using the combination of the terms '*radiation exposure*' and '*intradiscal ozone*' and/or '*ozone chemonucleolysis*' gave no pertinent result. To our knowledge, this could be the first study which compares radiation exposure to patients undergoing percutaneous intradiscal ozone therapy using different imaging modalities.

Getting to the point, we found quite low ED values in both groups of our series, probably due to the operators' experience in performing the percutaneous access to the intervertebral disc under fluoroscopy and, on the other side, to the use of radiation-sparing protocols for CT scanning.

In patients undergoing intradiscal ozone therapy using CT guidance, our mean DLP was 632.49mGy-cm (range 151.5–1699). The width of this range has different explanations. First of all the number of disc levels treated in the same session. Secondly, the ease of reaching the nucleus pulposus, sometimes hard to access because of spinal anatomy and/or bone modifications (spondilolysis, spondiloarthrosis and/or vertebral osteophytosis). In patients undergoing percutaneous intradiscal ozone therapy using fluoroscopy, our mean DAP was 11.63mGy*cm$^2$ (range 5.42–21.61). In this case, the range of values is narrower because the real-time fluoroscopic guidance allowed an uninterrupted monitoring of needle tip penetration, thus avoiding inaccurate trajectories.

Both techniques showed an optimal technical success (100%), which was proved by imaging the needle tip in the nucleus pulposus during all procedures. The clinical outcome of patients

undergoing fluoroscopy-guided and CT-guided procedures was compared using the ODI Questionnaire [10], with no significant difference in the mean ODI reduction (p-value = 0.41).

Overall, the clinical outcome was satisfactory: relief of pertinent symptoms was registered in 81.1% of patients. In terms of percentages, our results were similar to those described by Ezeldin in 2018 [27], who showed a 6-month follow up improvement of disability symptoms in 76% of patients, even if the results are slightly comparable due to the different follow up period considered. Differently, the mean reduction of ODI score registered in this series was of 15.64 points, which was significantly lower than the ODI decrease observed in our study (21.82±10.66). A paper by Gallucci [12] reported a success rate lower than ours (74% versus 81.3%) in patients treated with combined intradiscal and intraforaminal injections of oxygen-ozone, steroid, and local anesthesia. Another paper by Andreula [23] showed a satisfactory therapeutic outcome slightly lower than ours (78.3% versus 81.1%) in case of patients receiving an intradiscal (4mL) and periganglionic (8mL) injection of an oxygen-ozone mixture together with a periganglionic injection of corticosteroid and anesthetic, whereas the outcome was poorer if the solution of corticosteroid and anesthetic was waived (70.3%). In our series, we did not distinguish different types of disc herniations, as proposed in 2008 by Muto et al. [28], who described success rates of 75%-80% for soft disc herniation, 70% for multiple-disc herniations and 55% for failed back surgery syndrome. Thereby, our overall success rate of 81.1% was slightly superior to what registered by Muto for soft disc herniation, and definitely better than what described for multiple-disc herniations and failed back surgery syndrome. Probably, this fact can be partially explained with different recruitment criteria.

Eventually, it is to state that it was not possible to properly compare our results with previously published papers using a clinical outcome scoring other than the Oswestry Disability Questionnaire, such as Oder et al. [29] and Lu et al. [30]. The first reported successful treatment in 620 subjects with reduction of pain measured by means of VAS score with excellent results in one third of the patients (reduction from 8 to <3). In a different way, Lu et al. referred the therapeutic outcomes according to the modified Macnabcriteria [31] (excellent efficacy in 63.8% of patients, good/fair in 27.6%, poor in 8.6%), with total effective rate (excellent/good/fair) of 91.4%.

Also, our results showed a clinical outcome not inferior to what described in previously published papers describing other percutaneous disc decompression methods as well as surgical treatments [32–34], but further research comparing intradiscal ozone therapy and different discal treatments are needed to confirm this point.

The primary limitations of this study are the retrospective design and the single center nature. A larger sample size would add additional power to our results. Another limitation is the lack of indisputable conversion coefficients to compare the radiation dose for different guidance techniques (fluoroscopy and conventional CT). Eventually, a study standardized for the choice of the guidance technique would be desirable in order to avoid any statistical bias.

Contrary, a potential strength of this paper is the attention paid on radiation exposure for patients undergoing percutaneous intradiscal ozone therapy, which is a timely and hot topic in radiology, in our opinion.

In conclusion, fluoroscopic guidance for percutaneous ozone chemonucleolysis of symptomatic disc herniation is safe, highly effective, technically sound and clinically successful as much as conventional CT, with reduced radiation dose administered to patients.

## Author Contributions

**Conceptualization:** Francesco Somma, Vincenzo D'Agostino, Carmine Sicignano, Gianvito Pace, Giuseppe Sarti, Giovanna Pezzullo.

**Data curation:** Francesco Somma, Stefania Tamburrini, Fabrizio Fasano, Gianluca Gatta, Ferdinando Caranci.

**Formal analysis:** Francesco Somma, Carmine Sicignano, Fabrizio Fasano, Silvio Peluso, Giuseppe Sarti.

**Investigation:** Francesco Somma, Carmine Sicignano.

**Methodology:** Alberto Negro, Valeria Piscitelli, Stefania Tamburrini, Carmine Sicignano, Giuseppe Sarti, Giuseppe Maria Ernesto La Tessa.

**Project administration:** Alberto Negro, Giuseppe Maria Ernesto La Tessa.

**Resources:** Vincenzo D'Agostino, Giuseppe Maria Ernesto La Tessa.

**Software:** Stefania Tamburrini.

**Supervision:** Francesco Somma, Alberto Negro, Valeria Piscitelli, Silvio Peluso, Gianvito Pace.

**Validation:** Alessandro Villa, Gianvito Pace, Gianluca Gatta.

**Visualization:** Alessandro Villa, Gianvito Pace, Gianluca Gatta.

**Writing – original draft:** Francesco Somma.

**Writing – review & editing:** Francesco Somma, Vincenzo D'Agostino, Alberto Negro, Stefania Tamburrini, Fabrizio Fasano, Alessandro Villa, Giovanna Pezzullo, Ferdinando Caranci.

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
