## [Decision Letter · Decision Letter 0]

26 Jan 2022

PONE-D-21-30975Radiation exposure and Clinical Outcome in patients undergoing percutaneous intradiscal ozone therapy for disc herniation: Fluoroscopic versus Conventional CT guidance.PLOS ONE

Dear Dr. Somma,

Thank you for submitting your manuscript to PLOS ONE. After careful consideration, we feel that it has merit but does not fully meet PLOS ONE’s publication criteria as it currently stands. Therefore, we invite you to submit a revised version of the manuscript that addresses the points raised during the review process. Please address the comments from all reviewers adequately especially reviewer #1 to improve the quality of your manuscript. All the best for your revised manuscript.

We look forward to receiving your revised manuscript.

Kind regards,

Suhairul Hashim, PhD

Academic Editor

PLOS ONE

Journal Requirements:

2. Please ensure that you refer to Figure 1 and 2 in your text as, if accepted, production will need this reference to link the reader to the figure.

Reviewers' comments:

Reviewer's Responses to Questions

**Comments to the Author**

1. Is the manuscript technically sound, and do the data support the conclusions?

Reviewer #1: Yes

Reviewer #2: Yes

Reviewer #3: Partly

2. Has the statistical analysis been performed appropriately and rigorously? 

Reviewer #1: Yes

Reviewer #2: Yes

Reviewer #3: Yes

3. Have the authors made all data underlying the findings in their manuscript fully available?

Reviewer #1: Yes

Reviewer #2: Yes

Reviewer #3: Yes

4. Is the manuscript presented in an intelligible fashion and written in standard English?

Reviewer #1: Yes

Reviewer #2: Yes

Reviewer #3: Yes

5. Review Comments to the Author

Reviewer #1: Dear Editors,

Dear Authors,

Thank you for giving me the opportunity to review the article you submitted with the title "Radiation exposure and clinical outcome in patients undergoing percutaneous intradiscal ozone therapy for disc herniation: fluoroscopic versus conventional CT guidance".

The topic of the manuscript refers to the socially significant problem of low back pain and the possibilities for its treatment with minimally invasive approaches. The authors compare radiation exposure and results of intradiscal ozone administration under fluoroscopic and CT control.

The study presents data of primary scientific research and I found that the reported results have not been published elsewhere. The research meets all applicable standards for the ethics and research integrity. The article is presented in an intelligible fashion and adheres to appropriate reporting guidelines. Summary of results is given concisely in three tables and are clearly supported by accurate and good quality images. Conclusions are appropriate and the results as well as the discussion supports them.

My comments regarding the manuscript are listed below:

- Intradiscal procedures are not widely accepted for degenerative disc disease. Here no rationality or indications for ozone treatment are listed. In the context of the current article submitted for publication, the clinical results should overcome the risks of ration exposure, in particular of CT-guided procedures.

- Levofloxacin is not considered as optimal perioperative antimicrobial prophylaxis. Furthermore, a 5-day treatment contradicts the principle for minimally invasive treatments with reduced risk for surgical site infections. It is not recommended in any related guideline alone and as a first line for preventions of surgical site infection.

- The statistical methods need more detailed description regarding homogeneity of the groups of patients. However, the statistical and further analysis are performed to a high technical standard.

- A major remark concerns the definition of “clinical success”. "Complete disappearance of symptoms" is inappropriate definition for clinical success. It contradicts the methodology to use the ODI measurement instrument, which is an accepted tool for that. Successful procedure, defined as reduction in the preoperative ODI values of at least 30% during follow-up, is acceptable. Note that the definition of “residual symptoms” correlate with the last.

- The discussion of the authors’ results in the sense of other studies is adequate and fair except the comparison to those reported by Ezeldin (2018). The judgement that their ODI score is better is not appropriate because the follow-up is different (1- versus 6-month).

- The statement that the results are not inferior to those obtained with other percutaneous disc decompression methods as well as surgical treatments is speculative. The comparison to other treatment modalities due to different selection criteria and indications implies bias.

- Many technical errors and need for corrections are marked in the attached pdf file.

In conclusion, despite my remarks and the inaccuracies found, the proposed study is substantiated and presented concisely. The stated aim to compare the radiation exposure and the clinical outcome of the discussed minimally invasive procedure under fluoroscopic and CT guidance is schieved. I would recommend the proposed article to be accepted for publication but after clearance of my comments. I would recommend the authors to pay more attention when preparing their manuscripts on methodology and punctuation.

Yours Sincerely,

Dr. Dilyan Ferdinandov

Reviewer #2: This article clearly shows the object of study. In particular, notions of technique with exposure dose, clinical successes and complications are correctly highlighted. The statistical analysis is correct and the number of the population is satisfactory. Well done.

Reviewer #3: The authors propose an original article comparing the technical, clinical, and dosimetric outcome of Ct and fluorscopy guided ozone nucleolysis.

The topic is interesting, and there are few previous reports addressing this issue.

There are some issues to address in the revision:

-INTRODUCTION: too long and not focusing enough on the research topic, that is the difference in imaging guidance. the main differences of floroscopy-guided and CT guided procedures should be introduced here.

- METHODS: the study design should be described more clearly, as it is not clear if it is a retrospective or prospective study. Inclusion and exclusiuon criteria for patient selkection should be stated.

6. PLOS authors have the option to publish the peer review history of their article (what does this mean?). If published, this will include your full peer review and any attached files.

Reviewer #1: **Yes: **Dr. Dilyan Ferdinandov, MD, MPH, PhD, FEBNS

Associated Professor in Neurosurgery, Medical University – Sofia

Senior Neurosurgeon, St. Ivan Rilski University Hospital

15 Acad. Ivan Geshov Blvd., 1431 Sofia, Bulgaria

+359 888 678 549

d.ferdinandov@medfac.acad.bg

Reviewer #2: No

Reviewer #3: No

---

## [Author Response · Author response to Decision Letter 0]

31 Jan 2022

RESPONSE TO REVIEWERS

Authors (A): First and foremost, the authors would like to express their gratitude to all reviewers for the appreciable work performed on the proposed manuscript. All suggestions have been expressly considered in the attached revision. Please find below our response to your highly valued recommendations. 

Reviewer (R) #1:

R: Dear Authors, thank you for giving me the opportunity to review the article you submitted with the title "Radiation exposure and clinical outcome in patients undergoing percutaneous intradiscal ozone therapy for disc herniation: fluoroscopic versus conventional CT guidance". The topic of the manuscript refers to the socially significant problem of low back pain and the possibilities for its treatment with minimally invasive approaches. The authors compare radiation exposure and results of intradiscal ozone administration under fluoroscopic and CT control. The study presents data of primary scientific research and I found that the reported results have not been published elsewhere. The research meets all applicable standards for the ethics and research integrity. The article is presented in an intelligible fashion and adheres to appropriate reporting guidelines. Summary of results is given concisely in three tables and are clearly supported by accurate and good quality images. Conclusions are appropriate and the results as well as the discussion supports them. My comments regarding the manuscript are listed below.

A: Dear dr. Ferdinandov, we are extremely glad for the recognition of the strengths of our work. With particular regard to the fact that the topic of our manuscript has never been published, previously. Thank you very much for spending your time in improving our paper.

R: Intradiscal procedures are not widely accepted for degenerative disc disease. Here no rationality or indications for ozone treatment are listed. In the context of the current article submitted for publication, the clinical results should overcome the risks of ration exposure, in particular of CT-guided procedures.

A: As stated in the Ethics section, the studied procedure is ordinarily performed in our institution, and is not considered experimental, thus meaning that the risk due to radiation exposure are not superior to the clinical relief in the vast majority of patients with no previous response to drugs, even in patients undergoing conventional CT guidance.

R: The statistical methods need more detailed description regarding homogeneity of the groups of patients. However, the statistical and further analysis are performed to a high technical standard.

A: In the beginning of the Result section, more detailed information regarding groups’ homogeneity were provided, as requested. In particular, the two groups were found homogeneous in terms of patient age, patient weight, level treated and side of disc herniation.

R: A major remark concerns the definition of “clinical success”. "Complete disappearance of symptoms" is inappropriate definition for clinical success. It contradicts the methodology to use the ODI measurement instrument, which is an accepted tool for that. Successful procedure, defined as reduction in the preoperative ODI values of at least 30% during follow-up, is acceptable. Note that the definition of “residual symptoms” correlate with the last.

A: As the reviewer properly denoted, in the field of topic clinical success is not the Complete disappearance of symptoms but the reduction in the preoperative ODI values of at least 30% during follow-up. Therefore, the erroneous definition was removed and the right one was properly highlighted in the Mat&Met section Outcome measure subsection. 

R: The discussion of the authors’ results in the sense of other studies is adequate and fair except the comparison to those reported by Ezeldin (2018). The judgement that their ODI score is better is not appropriate because the follow-up is different (1- versus 6-month).

A: According to the reviewer suggestion, the the following sentence was added to make the reader aware of different follow up time: “even if the results are slightly comparable due to the different follow up period considered”

R: The statement that the results are not inferior to those obtained with other percutaneous disc decompression methods as well as surgical treatments is speculative. The comparison to other treatment modalities due to different selection criteria and indications implies bias.

A: As suggested by the reviewer, we changed the sentence in a more cautious way, as reported below: “Also, our results showed a clinical outcome not inferior to what described in previously published papers describing other percutaneous disc decompression methods as well as surgical treatments, but further research comparing intradiscal ozone therapy and different discal treatments are needed to confirm this point”. 

R: Many technical errors and need for corrections are marked in the attached pdf file.

A: A deep editing and language revision has been performed.

Reviewer (R) #2:

R: This article clearly shows the object of study. In particular, notions of technique with exposure dose, clinical successes and complications are correctly highlighted. The statistical analysis is correct and the number of the population is satisfactory. Well done.

A: Thank you very much for your appreciable comments.

Reviewer (R) #3:

R: The authors propose an original article comparing the technical, clinical, and dosimetric outcome of CT and fluorscopy guided ozone nucleolysis. The topic is interesting, and there are few previous reports addressing this issue. There are some issues to address in the revision.

A: We would like to thank you for the valuable comments helping us to improve our manuscript.

R: INTRODUCTION: too long and not focusing enough on the research topic, that is the difference in imaging guidance. the main differences of floroscopy-guided and CT guided procedures should be introduced here.

A: According to the reviewer suggestion, the Discussion section was shortened and re-written in order to focus radiation exposure and protection.

R: METHODS: the study design should be described more clearly, as it is not clear if it is a retrospective or prospective study. Inclusion and exclusion criteria for patient selection should be stated.

A: In order to clear the study design we added the following text in the section Mat&Med subsection Patients Population: “All cases were retrospectively reviewed”. Inclusion and exclusion criteria were added.

In conclusion, all authors would like to thank Editors and Reviewers for their time and the expertise provided in the revision of our manuscript.

---

## [Decision Letter · Decision Letter 1]

17 Feb 2022

Radiation exposure and Clinical Outcome in patients undergoing percutaneous intradiscal ozone therapy for disc herniation: Fluoroscopic versus Conventional CT guidance.

PONE-D-21-30975R1

Dear Dr. Somma,

We’re pleased to inform you that your manuscript has been judged scientifically suitable for publication and will be formally accepted for publication once it meets all outstanding technical requirements.

Kind regards,

Suhairul Hashim, PhD

Academic Editor

PLOS ONE

Additional Editor Comments (optional):

Congratulations for your great efforts in publishing with PLOS ONE.

Reviewers' comments:

Reviewer's Responses to Questions

**Comments to the Author**

1. If the authors have adequately addressed your comments raised in a previous round of review and you feel that this manuscript is now acceptable for publication, you may indicate that here to bypass the “Comments to the Author” section, enter your conflict of interest statement in the “Confidential to Editor” section, and submit your "Accept" recommendation.

Reviewer #1: All comments have been addressed

Reviewer #2: All comments have been addressed

Reviewer #3: All comments have been addressed

2. Is the manuscript technically sound, and do the data support the conclusions?

Reviewer #1: Yes

Reviewer #2: Yes

Reviewer #3: Yes

3. Has the statistical analysis been performed appropriately and rigorously? 

Reviewer #1: Yes

Reviewer #2: Yes

Reviewer #3: Yes

4. Have the authors made all data underlying the findings in their manuscript fully available?

Reviewer #1: Yes

Reviewer #2: Yes

Reviewer #3: Yes

5. Is the manuscript presented in an intelligible fashion and written in standard English?

Reviewer #1: Yes

Reviewer #2: Yes

Reviewer #3: Yes

6. Review Comments to the Author

Reviewer #1: (No Response)

Reviewer #2: For revisioned version of this paper, acceptable revisions have been made. I confirm my positive opinion

Reviewer #3: The authors fully adressed all the questions raised in the previous revision.

In particular, according to my suggestion, the Discussion section was shortened and re-written in order to focus radiation exposure and protection.

In addition, they added the following text in the section Mat&Med subsection Patients Population: “All cases were retrospectively reviewed”. Inclusion and exclusion criteria were added.

7. PLOS authors have the option to publish the peer review history of their article (what does this mean?). If published, this will include your full peer review and any attached files.

Reviewer #1: **Yes: **Dilyan Ferdinandov, MD, MPH, PhD

Reviewer #2: No

Reviewer #3: No

---

## [Editor Report · Acceptance letter]

1 Mar 2022

PONE-D-21-30975R1 

Radiation exposure and Clinical Outcome in patients undergoing percutaneous intradiscal ozone therapy for disc herniation: Fluoroscopic versus Conventional CT guidance. 

Dear Dr. Somma:

I'm pleased to inform you that your manuscript has been deemed suitable for publication in PLOS ONE. Congratulations! Your manuscript is now with our production department. 

Kind regards, 

on behalf of

Dr. Suhairul Hashim 

Academic Editor

PLOS ONE